# Multifaceted Roles of CD5L in Infectious and Sterile Inflammation

**DOI:** 10.3390/ijms22084076

**Published:** 2021-04-15

**Authors:** Lidia Sanchez-Moral, Neus Ràfols, Clara Martori, Tony Paul, Érica Téllez, Maria-Rosa Sarrias

**Affiliations:** 1Innate Immunity Group, Health Research Institute Germans Trias i Pujol, Ctra Can Ruti camí de les escoles, s/n, 08916 Badalona, Spain; lsanchez@igtp.cat (L.S.-M.); nrafols@igtp.cat (N.R.); cmartori@igtp.cat (C.M.); tpaul@igtp.cat (T.P.); etellez@igtp.cat (É.T.); 2Network for Biomedical Research in Hepatic and Digestive Diseases (CIBERehd), 28029 Madrid, Spain

**Keywords:** AIM, Spα, macrophage, apoptosis, scavenger receptor cysteine-rich, CD36

## Abstract

CD5L, a protein expressed and secreted mainly by macrophages, is emerging as a critical immune effector. In addition to its well-defined function as an anti-apoptotic protein, research over the last decade has uncovered additional roles that range from pattern recognition to autophagy, cell polarization, and the regulation of lipid metabolism. By modulating all these processes, CD5L plays a key role in highly prevalent diseases that develop by either acute or chronic inflammation, including several infectious, metabolic, and autoimmune conditions. In this review, we summarize the current knowledge of CD5L and focus on the relevance of this protein during infection- and sterile-driven inflammatory pathogenesis, highlighting its divergent roles in the modulation of inflammation.

## 1. Introduction

Innate immunity relies on a network of receptors and cellular effectors able to recognize non-self and also damaged-self molecular patterns and trigger a response to protect the host. Pathogen-associated molecules (PAMPs), such as lipopolysaccharide (LPS) from *Escherichia coli*, lipids from the cell wall of *Mycobacterium tuberculosis*, and ssRNA from viral particles, are among the non-self-molecules that engage innate immunity. Likewise, oxidized low density lipoprotein (LDL), adenosine 5′-triphosphate (ATP), and hyaluronan fragments, among others, constitute danger or damage-associated molecular patterns (DAMPs) that are either modified or released from damaged or dying cells [1]. Subsequently, a series of well-coordinated proinflammatory programs, with accompanying migration of killer and phagocytic cells such as macrophages, are mounted to destroy the harm, remove cell debris, and eventually promote wound healing and boost adaptive immunity.

However, the inflammatory response may act as a double-edged sword. PAMP- or DAMP-mediated signaling may exacerbate the inflammatory state in a disproportional matter due to the genetic makeup of the host and/or persistence of the pathogen or damage, thereby leading to dysregulation and additional tissue damage. This exacerbation may cause systemic acute or chronic inflammatory conditions, leading to sepsis, infarction, or abnormal tissue remodeling (fibrosis) among others, thus seriously compromising the health of the host [2].

Among the repertoire of innate immune effectors, CD5L (CD5-like molecule) is a secreted protein produced mostly by macrophages. It belongs to the scavenger receptor cysteine-rich (SRCR) family of proteins. In the human, it consists of 347 amino acids, with a secretory signal sequence of 19 hydrophobic amino acids that are absent in the mature protein [3,4], followed by three SRCR domains, each approximately 100 amino acids in length with sialic acid content at a potential region of O-linked glycosylation between SRCR domains 1 and 2 [4] (Figure 1). With 68% sequence identity, mouse CD5L undergoes different post-translational modifications, its larger molecular weight (55 kDa) being explained by the presence of three putative N-glycosylation sites, two of which were verified to bind to N-glycans and to affect its secretion and function [5] (Figure 1).

CD36 is considered the main cell-surface receptor for CD5L. The CD36–CD5L interaction was initially described by Kurokawa and colleagues when they showed that adipocytes internalized CD5L in a CD36-dependent manner [6]. In vitro experiments revealed that the presence of CD36-neutralizing antibodies decreased the uptake of recombinant CD5L (rCD5L) by adipocytes—a finding that was further supported by experiments performed in vivo, where the uptake of rCD5L by CD36-deficient mice was found to be lower when compared to wild-type (WT) mice [6]. CD36 is a transmembrane glycoprotein expressed by a wide range of cell types, including macrophages, dendritic cells, endothelial cells, adipocytes, hepatocytes, and cardiac and epithelial cells [7], and as such, it allows CD5L to target multiple types of tissue. However, CD5L has also been described to target CD36 non-expressing cells such as natural killer T (NKT) cells and thymocytes [8,9], and to bind to alternative receptors, such as kidney injury molecule (KIM-1) in kidney epithelial cells [10] (see below). These observations, therefore, indicate there are additional receptors for CD5L.

CD5L is a pleiotropic protein that exerts multiple activities, the inhibition of apoptosis being the most unifying and conserved function. In this regard, CD5L supports the survival of macrophages and other cell types when challenged with various apoptosis-inducing insults of infectious origin and chemical compounds [8,9]. Additional activities, ranging from pattern recognition to the modulation of inflammatory responses, autophagy, cell polarization, and the control of lipid metabolism, make CD5L a key element in the maintenance of tissue homeostasis. Through modulation of all these processes, CD5L plays an important role in both acute and chronic inflammatory processes, including in several infectious, metabolic, and autoimmune diseases (Figure 2). However, research on CD5L has provided evidence of divergent or even opposite outcomes of its involvement in certain processes, mostly related to its role in either promoting or attenuating inflammatory responses. In this review, we summarize the different roles exerted by CD5L in the modulation of leukocyte inflammatory responses to PAMPs and DAMPs and highlight the divergence regarding the beneficial vs. detrimental effects of CD5L described to date.

## 2. CD5L in Infection

Given the multifunctional character of CD5L, it is essential to decipher its involvement in bacterial, viral, or parasitic infections. It is not yet clear whether this protein supports or hinders pathogens since distinct outcomes have been reported. However, certain functions, such as its capacity to bind to pathogen-related structures and its antiapoptotic role in macrophages during the course of infection, have been widely described. These and other roles of CD5L in infection are summarized below.

### 2.1. Binding to Pathogens

The recognition of pathogen structures is a key step in engaging innate immune responses to increase the host’s ability to respond to infection. In this regard, SRCR domains are characterized by having broad ligand binding capacity. Accordingly, SRCR receptors MARCO [11], DMBT1/SAG/gp340/SALSA [12], CD6 [13], CD163 [14], CD5 [15], and SRA1 [16] are able to interact with multiple ligands. Likewise, CD5L can bind to Gram-negative and -positive bacteria [17,18] and fungi [17]. The presence of multiple binding sites in each of the three highly homologous SRCR tandem repeats could explain the additional ability of CD5L to aggregate these microorganisms [17,18]. It has been proposed that the aggregation capacity of CD5L may prevent these pathogens from invading cells and contribute to their elimination.

### 2.2. Phagocytosis and Bacterial Killing

In addition to its pathogen-binding properties, CD5L participates in antimicrobial responses, such as phagocytosis, which it enhances. This phenomenon has been observed for latex beads [19], cellular debris [10] and apoptotic cells [20] and also for infectious agents. In the context of *Staphylococcus aureus*-induced pneumonia, pre-incubation of isolated mouse alveolar macrophages and neutrophils with recombinant CD5L (rCD5L) for 1 h triggered an increase in bacterial phagocytosis. However, this increase was not accompanied by greater intracellular bacterial killing [21]. Moreover, in a mouse model of *Corynebacterium parvum*-induced hepatic granulomas, macrophage phagocytosis of bacilli was also impaired in CD5L-deficient animals when compared to WT counterparts [9]. This finding is in contrast to that of a more recent study by Sanjurjo et al., where the incubation of human peripheral blood monocytes with rCD5L for 24 h did not modify the phagocytosis of microspheres, *Escherichia coli* or *S. aureus* particles [20]. The activity of mouse and human CD5L may differ with respect to the promotion of phagocytosis. Moreover, differences in the experimental design used in these studies may be a decisive factor for the outcome.

In the setting of intracellular bacterial infection, early in vitro experiments showed that overexpression of CD5L in RAW macrophages promoted antimicrobial activity against *Listeria monocytogenes* [22]. Later, CD5L was shown to inhibit the growth of another intracellular bacteria, namely *M. tuberculosis*, in THP1 macrophages [23]. In vitro, CD5L macrophage expression peaked in the early phase of infection, triggering the synthesis of vitamin D-dependent antimicrobial peptides and subsequent autophagic killing of mycobacteria. These findings revealed that CD5L participates in the host response to intracellular bacteria and provided novel insights into the intracellular mechanisms activated by this protein [23]. Autophagy is a highly conserved cellular degradation process that serves to recycle obsolete damaged or superfluous cell components into basic biomolecules [24]. Moreover, in the macrophage, autophagy also functions as a key mechanism against bacterial infections because it directs intracellular pathogens to lysosomes for degradation while controlling inflammation to limit damage to the host [25]. Accordingly, in a more recent study, in both THP1 macrophages and peripheral blood monocytes, CD5L increased cellular LC3-II content, LC3 puncta, and also LC3-LysoTracker Red colocalization, all well-known markers of autophagy. Furthermore, electron microscopy experiments indicated that THP1 macrophages overexpressing CD5L had a higher number of cytoplasmic autophagosomes. Silencing experiments suggested that the scavenger receptor CD36 was required for CD5L-induced autophagy, thus revealing a new role for the CD36–CD5L axis in the induction of macrophage autophagy and the regulation of cellular homeostasis [26].

### 2.3. Inflammatory Responses In Vitro

CD5L also binds to free PAMPs released by bacteria, namely LPS and lipoteichoic acid (LTA) [18] as well as peptidoglycan (PGN), mannan, β-d-glucan, and zymosan [17], and inhibits the proinflammatory responses of monocytes. In addition, rCD5L inhibited monocyte TNF [17,18] and IL-1β production while enhancing IL-10 secretion [26] upon stimulation with LPS and Pam3CSK4, which are molecules originated from Gram-negative and Gram-positive bacteria, respectively. Likewise, mouse splenocyte cytokine release induced by LPS, as well as by fungal β-glucan and PGN, was inhibited in the presence of rCD5L [17]. This CD5L anti-inflammatory action is thought to be mediated, at least in part, by enhancing the autophagic machinery of macrophages [26].

In the course of an infection, tissue macrophages will respond to microenvironmental stimuli by differentiating into specific phenotypes, which allow them to acquire the necessary functions: from proinflammatory, microbicidal, and subsequent tissue-damaging (referred to as M1 or classically activated macrophages) to anti-inflammatory, immunosuppressive, and wound-healing (referred to as M2 or alternatively activated) [27]. In this regard, and consistent with an anti-inflammatory role for CD5L, autophagy induction by this protein was shown to drive macrophages toward an M2 phenotype, similar to that induced by IL-10. Accordingly, the culture of peripheral blood monocytes with rCD5L for 24–72 h induced expression of CD136, MERTK, CD36, and VEGF, as IL-10 did. These CD5L-polarized macrophages secreted lower levels of inflammatory mediators (TNF-α, IL-1β, and IL-6) in response to LPS and showed an increased rate of phagocytic clearance of apoptotic cells. These studies provided the first evidence that CD5L influences macrophage plasticity [20].

Other studies also identified CD5L as a switch that regulates the functional state of helper CD4+ T (Th)17 cells and restrains their pathogenicity [28,29]. Th17 cells have diverse functions. They play a critical role in host defense against pathogens, they are present at tissue inflammation sites, and they contribute to the pathogenesis of several diseases [30]. Wang and colleagues reported that Th17 cells differentiated in vitro under pathogenic conditions (exposed to IL-23) lacked CD5L expression, whereas those differentiated under non-pathogenic conditions predominantly expressed CD5L [28]. Interestingly, although the gene was expressed only in non-pathogenic cells, their expression level correlated with the proinflammatory module, acting as a negative regulator. CD5L mediated its effect on Th17 cell pathogenicity by modulating the intracellular lipidome, altering fatty acid composition, and restricting cholesterol biosynthesis and, thus, ligand availability for Rorγt, the master transcription factor of these cells. This effect of CD5L on lipid metabolism reduces Rorγt binding to the pathogenic IL-23r and IL-17 regions and increases its binding at the protective IL-10 region [28]. Additional studies revealed that the p19 subunit of IL-23, a cytokine that plays a critical role in the differentiation and maintenance of pathogenic Th17 cells, is secreted in the culture supernatant of activated CD4+ T cells and associates with CD5L to form a p19/CD5L heterodimer, which subsequently activates STAT5 and enhances differentiation into granulocyte macrophage colony-stimulating factor (GM-CSF)-producing CD4+ T cells [29].

### 2.4. Protection from Apoptosis

CD5L was first identified as a macrophage-secreted protein able to bind to the surface of leukocytes [3,31], and to protect thymocytes and J774A.1 monocytes from apoptosis [8], hence its original names Spα (soluble protein alpha) [3,31], and apoptosis inhibitor expressed by macrophages (AIM) [8], respectively. In the context of infection, CD5L contributes to protecting macrophages against apoptosis induced by various microorganisms, namely *Bacillus anthracis*, *E. coli*, *Salmonella typhimurium*, and *L. monocytogenes* [22,32,33].

Knowledge of the anti-apoptotic role of CD5L in infections arose from the study of nuclear receptor transcription factors LXR/RXR in infection and inflammation. LXR/RXR are cholesterol-sensing receptors that have emerged as key regulators of lipidic metabolism and transport as well as inflammatory responses, thus providing an important link between metabolism and immunity. Several studies have shown that CD5L expression is induced through LXR/RXR activation by its natural ligands, either exogenous (oxLDL) or endogenous (25-hydroxycholesterol, 25HC; produced by cholesterol-25-hydroxylase enzyme, Ch25h), or by synthetic LXR/RXR ligands (T1317, 9cRA, GW3965) [22,32,33,34].

Two distinct transcription factors highly expressed in macrophages and upregulated by activated LXR/RXR also participate in CD5L regulation: sterol regulatory element-binding protein (SREBP-1), which connects lipidic metabolism to diverse physiologic responses [35,36] and MafB, which induces myelomonocytic differentiation and has recently been associated with atherogenesis [37]. In addition, SREBP-1 was demonstrated to induce the transcription of the *CD5L* gene, leading to a reduction of macrophage apoptosis. This phenomenon was observed in vitro during mouse bone marrow derived macrophages (BMDM) response to *S. aureus* pore-forming alpha-toxin, as well as to other bacterial toxins such as streptolysin O and LPS [36].

Little is known about the intracellular mechanisms underlying CD5L regulation of apoptosis. In vitro experiments showed that overexpression of CD5L in RAW macrophages prevented *L. monocytogenes*-induced caspase-3 activity [22]. In a later study involving in vitro experiments targeting CD5L expression by siRNA in *L. monocytogenes*-infected macrophage-like ZBM2 cells, CD5L enhancement of macrophage survival was attributed in part to the inhibition of caspase-1 activation. This finding suggested a blocking of pyroptosis, an inflammatory form of cell death [33]. Further studies are needed to unravel the cellular mechanisms activated by CD5L to protect macrophages from apoptosis.

### 2.5. Overall Action of CD5L during In Vivo Infection: Friend or Foe?

There is currently no consensus on whether the specific action of CD5L upon infection is beneficial or detrimental for the host. Actually, contradictory results have been observed under different settings, as explained below.

In a study on mice lacking LXR, these animals became highly susceptible to *L. monocytogenes* infection [22]. This finding was attributed to defective bacterial clearance as well as accelerated macrophage apoptosis due to loss of CD5L expression. However, these results are in apparent contradiction with another experimental model of *L. monocytogenes* infection, in which cholesterol-25-hydroxylase (Ch25h), the enzyme that synthesizes the LXR natural ligand 25-hydroxycholesterol (25HC), was transiently overexpressed [33]. In this scenario, mice were more susceptible to infection and showed higher bacterial loads in liver and spleen, which correlated with increased bacterial content in macrophages infected in vitro. The opposite outcomes of silencing LXR vs. transient overexpression of Ch25h may be due, in part, to the influence of 25HC on immune responses, independently of LXRs [38]. In addition, Ch25h overexpression at the time of infection enhanced the survival of *L. monocytogenes*-infected macrophages and intensified the disease. This finding suggests that constitutive vs. transient changes in macrophage apoptosis may have opposite outcomes [38].

In a second setting, using a mouse model of experimental sepsis induced by cecal ligation puncture (CLP), activation of the LXR pathway with recombinant growth differentiation factor 3 (rGDF3) improved mice survival along with significant reductions in bacterial load, plasma pro-inflammatory cytokine levels, and organ damage. This correlated with CD5L being the most significantly upregulated gene in rGDF3-treated macrophages [39]. These observations seem to contradict what was observed by Gao et al. [40], where blockade of CD5L with an intraperitoneally administered anti-CD5L antibody immediately after CLP resulted in an improvement of the disease as a result of a decrease in inflammation, namely a reduction of innate cell infiltration and of proinflammatory cytokine release. Likewise, intraperitoneal administration of rCD5L to septic mice immediately after CLP reduced their survival when compared to PBS-injected mice in the same condition [40].

Apparently contradictory conclusions on the role of CD5L have also been reached in additional in vivo inflammatory settings. Using a mouse model of zymosan-induced peritonitis, developed by the daily intraperitoneal injection with zymosan for 5 days, peritonitis was more severe in CD5L^−/−^ than in WT mice. Moreover, the administration of rCD5L to CD5L^−/−^ mice reduced inflammation in comparison with the non-treatment group. The anti-inflammatory effects of CD5L were attributed to an enhancement of necrotic tissue clearance rather than to antimicrobial activity. In this regard, peritoneal cells isolated from CD5L^−/−^ mice showed reduced uptake of fluorescently labeled dead cell debris, and coating debris with rCD5L increased the phagocytic capacity of macrophages [41]. In contrast, the resolution of inflammation after LPS-induced lung injury in mice was accelerated in CD5L^−/−^ mice when compared to WT mice, suggesting a role for CD5L in maintaining inflammatory responses [42].

Overall, these studies have revealed new intersection points between metabolic and inflammatory pathways in which CD5L plays a central role. Moreover, a variety of functions have been attributed to CD5L during the course of an infection, from the recognition and aggregation of infectious agents to the increase in the phagocytic capacity of macrophages, among other antimicrobial properties (Table 1). Nonetheless, no unifying role for this protein in the fight against infectious diseases has been defined. This is probably in part due to the lack of research and the diversity of the experimental designs used, but also because of the multifunctional character of the protein and the delicate balance present between inflammation and resolution of infection.

## 3. CD5L in Sterile Inflammation-Driven Pathogenesis

CD5L participates in several chronic and acute processes driven by sterile inflammation. Through its ability to bind to specific DAMPs, promote phagocytosis, block apoptosis, and control inflammatory cell migration and polarization, CD5L is a critical player in several pathological conditions, which are recapitulated below (Figure 2 and Table 1). In addition, a few studies propose that CD5L plays a role in cancer, but this condition is beyond the scope of the present review.

### 3.1. Chronic Inflammation

#### 3.1.1. Atherosclerosis

Hyperlipidemia and inflammation are major contributors to the accumulation of fat deposits and cellular debris within the arterial wall, a process that leads to atherosclerosis. LDL is a major extracellular carrier of cholesterol and, as such, it plays key physiologic roles, distributing cholesterol to peripheral tissues through the circulatory system. However, under conditions of hyperlipidemia, specific components of LDL become oxidized (oxLDL) or otherwise modified, and these modifications substantially alter the function of these molecules. Modified LDL particles act as a chemoattractant for monocytes, enhancing their migration and differentiation into macrophages and causing alterations to the endothelium. Moreover, macrophages uptake modified LDL through scavenger receptors, thereby turning into lipid-rich foam cells. Increasing amounts of these cells in the artery wall and subsequent proinflammatory environment lead to the development of atherosclerotic lesions which can obstruct the arterial lumen and/or eventually break and generate a thrombus, leading to myocardial infarction or stroke [53,54]. In this context, we demonstrated that CD5L binds to oxLDL and promotes CD36-mediated oxLDL uptake and thus serves as a soluble chaperone for oxLDL internalization [34]. Through this mechanism, CD5L may contribute to increasing the formation of macrophage foam cells. Moreover, in humans and mice, foam cells within atherosclerotic lesions express high levels of CD5L, which enhances the survival of macrophages within the artery wall, facilitating atherogenesis [43]. This was observed in a study comparing mice lacking both CD5L and LDL receptor (LDLR) versus those only lacking LDLR, fed with a high-fat, high-cholesterol diet. Under these settings, the development of atherosclerotic lesions was markedly reduced in double deficient CD5L and LDLR mice in comparison to LDLR-deficient mice [43]. Accordingly, a study reported that MafB, a transcription factor that directly regulates CD5L expression, also contributes to the progression of atherosclerosis by inhibiting foam-cell apoptosis [37].

#### 3.1.2. Obesity-Associated Inflammatory Diseases

Obesity is linked to insulin resistance—a condition that triggers and/or accelerates a variety of metabolic disorders such as type 2 diabetes, cardiovascular diseases, and fatty liver. Insulin resistance in obese individuals is caused, in part, by chronic, low-grade inflammation in adipose tissue [55]. The innate immune system is largely responsible for this subclinical state of inflammation. In the adipose tissue of obese subjects, fatty acids may activate toll-like receptors (TLRs) present on adipocytes, triggering the production of inflammatory mediators and promoting the infiltration of inflammatory macrophages, thus enhancing the chronic state of low-grade inflammation [56,57].

In studies analyzing its possible role in inflammation and obesity, CD5L internalized into adipocytes through CD36, and once in the cytosol, induced lipolysis [49]. This was achieved by binding to fatty acid synthase (FASN), a metabolic enzyme that catalyzes the synthesis of saturated fatty acids, such as palmitate, from acetyl-CoA and malonyl-CoA precursors. By binding to FASN, CD5L inhibited its enzymatic activity, thereby lowering the amount of saturated fatty acids in the adipocyte [6] and reducing the transcriptional activity of peroxisome proliferator-activated receptor (PPARγ). PPARγ inhibition led to reduced expression of two proteins essential for triacylglycerol storage, namely fat-specific protein 27 (FSP27) and perilipin [49]. These events resulted in decreased lipid droplet size, lower numbers of mature adipocytes, and decreased weight and fat mass induced by high-fat diet in mice [6,49]. In addition, CD5L-induced lipolysis caused free fatty acid efflux from adipose cells, thereby activating TLR4 inflammatory signaling in bystander adipocytes and inducing the recruitment of inflammatory macrophages [46]. Supporting these observations, obese CD5L-deficient mice had lower infiltration of inflammatory macrophages in adipose tissue, which prevented the progression of obesity-associated inflammation both locally and systemically. Moreover, the absence of CD5L appeared to prevent insulin resistance in obese mice [46]. Based on these results, the authors proposed the regulation of CD5L levels as a potential therapeutic strategy to treat inflammatory diseases associated with obesity, such as metabolic syndrome [46].

#### 3.1.3. Liver Fibrosis

Hepatic fibrosis is a complex mechanism in which chronic injury of any etiology, such as viral infection, alcohol consumption, or steatosis, induces recurrent inflammation and extracellular matrix (ECM) accumulation. If the problem persists, it may progress to cirrhotic disease, where the liver parenchyma and vascular architecture become distorted, causing severe morbidity and mortality [58]. Regarding the role of CD5L in hepatic fibrosis, studies by our group using an in vivo model of moderate fibrosis induced by the hepatotoxic compound carbon tetrachloride (CCl4) demonstrated that administration of rCD5L had a protective effect on liver injury and lowered the inflammatory response [45]. In brief, rCD5L administration attenuated CCl4-induced injury, as determined by reduced serum transaminase levels. The extent of liver fibrosis was also reduced, as observed by lower collagen content. In this regard, rCD5L had a direct effect on primary human hepatic stellate cells—the main hepatic cellular source of fibrotic deposition—through enhanced SMAD7 expression and repression of transforming growth factor beta (TGFβ) signaling. Furthermore, rCD5L prevented immune cell infiltration and promoted a phenotypic shift in monocytes from LyC6hi to LyC6low in CCl4-treated mice. LyC6low monocytes have been proposed to orchestrate the regression of liver fibrosis and, interestingly, to overexpress Cd5l [45]. On the basis of these results, CD5L emerges as a key player in fibrosis in the setting of chronic liver disease.

#### 3.1.4. IgA Nephropathy

IgA nephropathy (IgAN) is a leading cause of chronic kidney disease and renal failure worldwide. IgAN progression is influenced by the confluence of genetic and environmental factors and it is characterized by the presence of glomerular deposition of immune complexes, comprised of IgA1, together with IgG, IgM, and C3 of the complement system [59]. These immune deposits cause persistent inflammatory immune responses, which associate with enhanced macrophage infiltration and matrix expansion, which together lead to further glomerular damage [47]. In blood, CD5L binds to IgM pentamers, which stabilize and protect it from renal excretion [50,51]. Using a gddY mouse model that spontaneously recapitulates the pathogenesis observed in human IgAN, it was observed that CD5L derived from the circulation accumulated at glomeruli and co-localized with IgA deposits. Interestingly, although glomerular IgA deposition was also found in CD5L-deficient gddY mice, these animals were protected against renal injury due to the absence of IgG, IgM and C3 deposition. Thus, CD5L is essential for binding of IgM and/or IgG to previously deposited IgA to promote leukocyte infiltration, glomerular and interstitial fibrosis, extracellular matrix expansion, and increased expression of the proinflammatory genes Il-1β, IL-6, and TNF-α [47], which together enhance chronic renal damage.

#### 3.1.5. Autoimmune Encephalomyelitis

CD5L influences the plasticity of Th17 cells, which play a role in autoimmunity [28,30]. Wang et al. confirmed the importance of CD5L in vivo by showing that it is expressed in non-pathogenic Th17 cells but not in pathogenic cells from mice with experimental autoimmune encephalomyelitis (EAE). Furthermore, loss of CD5L converted non-pathogenic Th17 cells into pathogenic cells that induced autoimmunity [28]. Moreover, additional studies have revealed that the p19 subunit of IL-23 binds to CD5L and, during the course of EAE, serum levels of p19/CD5L, but not CD5L alone, are highly correlated with clinical symptoms. Both CD4+ T-cell-specific conditional p19-deficient mice and complete CD5L-deficient mice showed significantly alleviated EAE symptoms with a reduced frequency of GM-CSF + CD4+ T cells. All together, these studies reveal the p19/CD5L complex as a potential novel heterodimeric cytokine that contributes to EAE development [29].

#### 3.1.6. Macular Degeneration

Considered the leading cause of visual impairment and loss, age-related macular degeneration (AMD) is a disease that affects the macular region of the retina [60]. The pathology is characterized by drusen and abnormalities of the retinal pigment epithelium (RPE). Although AMD is not considered a classic autoimmune disease, both inflammation and the immune system play a crucial role in the pathogenesis. Studies addressing antibody recognition of autoantigens revealed that CD5L is physiologically expressed in the RPE and human retinal microglial cells, and have identified CD5L as an autoantigen involved in self-reactivity in AMD [52]. Serum from AMD patients presented higher circulating levels of CD5L when compared to control subjects, as well as higher levels of circulating autoantibodies that bound to CD5L [52]. Given that oxLDL is a component of drusen, the authors proposed that CD5L, together with CD36, plays a physiological role in the clearance of oxLDL from the retinal epithelium environment and limits the formation of drusen, while autoantibodies that bind to CD5L enhance drusen biogenesis by impairing oxLDL clearance. Autoantibody blockade of CD5L would also impair CD5L-mediated autophagy in AMD, thus promoting NLRP3 inflammasome activation and exacerbating AMD progression [61].

### 3.2. Acute Inflammation

#### 3.2.1. Myocardial Infarction

Myocardial infarction causes an inflammatory response to the release of endogenous molecules by necrotic cells and the extracellular matrix [62]. This response stimulates the production of inflammatory mediators and growth factors, sequentially inducing the recruitment of inflammatory cells, the clearance of injured tissue, angiogenesis, and the proliferation of fibroblasts, eventually resulting in scar formation and infarct healing. Dysregulation of these mechanisms may result in damage to cardiac myocytes and fibrosis, eventually leading to functional compromise of the heart. In this regard, the absence of CD5L reduced the inflammatory response and infarct size in an acute myocardial infarction mouse model [48]. This finding thus suggests that CD5L plays an important role in sustaining inflammation and the progression of cardiac remodeling. The beneficial effects of CD5L suppression were associated with a reduction of the inflammatory response driven by the TLR4 pathway. In particular, it was observed that NF-kB and IRAK-4 activation in the TLR-4/NF-kB pathway in myocardial tissue after coronary ligation was diminished in CD5L^−/−^ mice compared with WT mice. Moreover, the activities of MPO and iNOS were also decreased in CD5L^−/−^ mice after infarction, thereby suggesting reduced myocardial tissue injury due to the infiltration of neutrophils and subsequent production of reactive oxygen species (ROS) [48]. The authors suggested that the proinflammatory action of CD5L is due, in part, to the enhanced release of free fatty acids, which are known activators of the TLR4 pathway. However, modulation of TLR4 responses to additional DAMPs—such as HSP60, HMG1, and ECM—by CD5L could not be ruled out.

#### 3.2.2. Acute Kidney Injury

Using the same CD5L^−/−^ mice, Arai et al. examined the role of this protein in an experimental model of acute kidney injury (AKI) by ischemia reperfusion (IR) injury [10]. In the absence of CD5L, mice presented impaired recovery with enhanced levels of serum blood urea nitrogen and creatinine, two markers of renal dysfunction, as well as increased mortality at day 7 post-AKI. Interestingly, the absence of CD5L in these animals was not associated with increased apoptosis of tubular epithelial cells, thus eliminating a role for this protein in protecting cells from death in this setting. Likewise, the number of infiltrating macrophages appeared to be similar, suggesting no role for CD5L in macrophage recruitment. Instead, CD5L^−/−^ mice showed abrogated clearance of dead cells, which caused an increase in inflammation as a result of intraluminal obstruction by necrotic cell debris. In a series of in vitro experiments, the authors showed that CD5L acts as a chaperone to increase the engulfment of cellular debris by epithelial cells overexpressing the three phagocytosis receptors T-cell immunoglobulin mucin receptor 2 TIM-2, CD36, and KIM-1. In addition, they demonstrated that the role of CD5L in AKI is specifically mediated through its interaction with KIM-1. This type I membrane protein, also named HAVCR1 (hepatitis A virus cellular receptor 1) or TIM-1 (T-cell immunoglobulin mucin receptor 1) [63], acts as a receptor for phospholipids exposed in the membrane of apoptotic and dead cells, thereby mediating their internalization. Ischemia induces the overexpression of KIM-1 in proximal renal tubular epithelial cells, providing them with the capacity to transform into “semi-professional” phagocytic cells that remove and process apoptotic cells and necrotic tissue. Accordingly, treatment of mice with IR-induced AKI using rCD5L ameliorated renal pathology in WT and CD5L-deficient mice but not KIM-1-deficient counterparts [10]. These results provide important novel information on the inflammatory mechanisms induced by IR in kidney disease.

#### 3.2.3. Ischemia Reperfusion Injury in Liver

Ischemia reperfusion injury is also a key driver of tissue damage during liver surgery, in procedures such as resection and transplantation [64]. It is a cascade of pathophysiological events in which cellular injury by oxygen deprivation induces the overproduction of ROS and hepatocyte apoptosis. In this context, a recent study using an in vitro model of IR in cultured hepatocytes showed that addition of rCD5L protected these cells from IR-induced apoptosis by reducing caspase-3/-7 and caspase-8 activity [44]. Moreover, addition of rCD5L to the cultures lowered the production of intracellular ROS. In these assays, targeting of CD36 and ubiquitin-like modifier-activating enzyme ATG7—a key protein in autophagy signaling pathway—limited CD5L activity [44]. The results, therefore, suggest that activation of autophagy through CD36 receptor is involved in the protective effects exerted by CD5L, a notion consistent with the results observed in [26].

## 4. Concluding Remarks

The first reports on CD5L defined it as an inhibitor of apoptosis. Nevertheless, research during the two last decades has unveiled countless additional roles, thus placing CD5L as an essential component in the maintenance of tissue homeostasis and during inflammatory processes. Through its multiple activities, mostly in leukocytes but also in other cell types, CD5L modulates the progression of high prevalence disorders, ranging from infectious to sterile inflammation-driven diseases. However, to date, studies have failed to decipher the mechanisms underlying its divergent roles (advantageous or detrimental) in the modulation of inflammation. A greater understanding of its multiple functions will be key for the development of anti-/pro-CD5L compounds, which will provide the basis for novel therapies in these pathological settings.

## Figures and Tables

**Figure 1 ijms-22-04076-f001:**
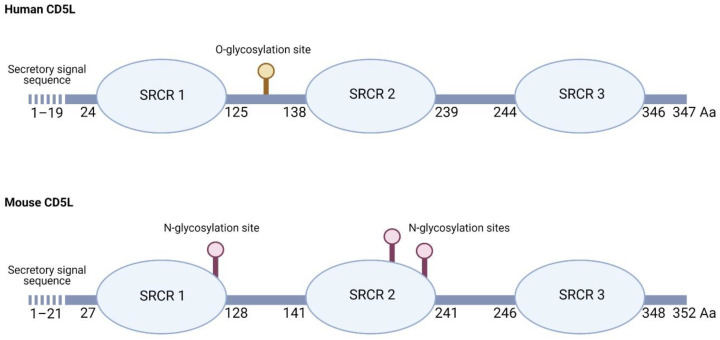
Schematic diagram of human and mouse CD5L. The scavenger receptor cysteine-rich (SRCR) domain structure, secretory signal peptide sequence, O- and N-glycosylation sites, and the amino acid (Aa) number are indicated in each sequence, according to [3,4,5] and the Uniprot Consortium (figure created with Biorender.com).

**Figure 2 ijms-22-04076-f002:**
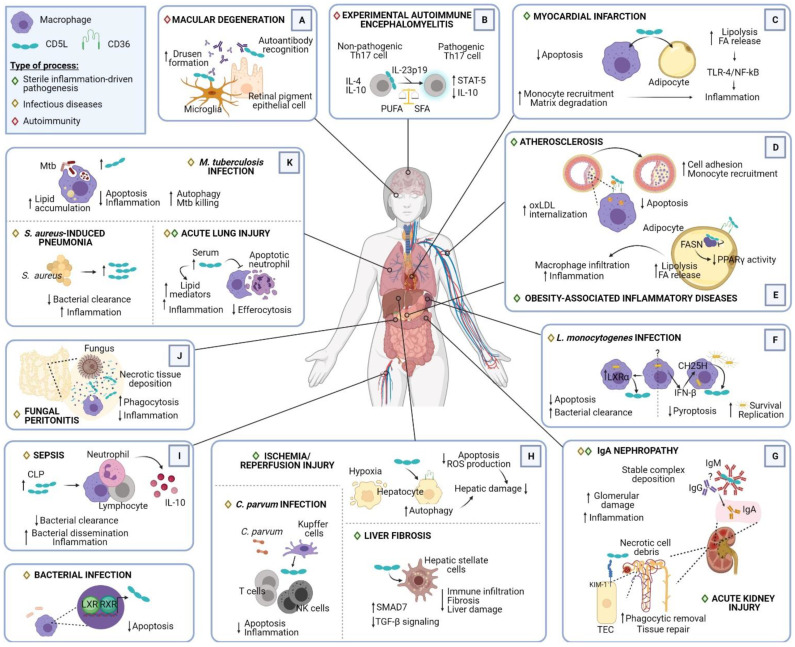
Multiple roles of CD5L in pathogenesis. Representative illustration summarizing the different scenarios in which CD5L plays a role, including its source, target cells, interacting proteins, and main outcomes. The processes have been classified into sterile inflammation-driven pathogenesis, infectious diseases, and autoimmunity, and have been grouped according to the affected region: (**A**) eye; (**B**) brain; (**C**) heart; (**D**) artery; (**E**) adipose tissue; (**F**) spleen; (**G**) kidney; (**H**) liver; (**I**) circulation; (**J**) peritoneum; (**K**) lung. ↓, decreased; ↑, increased. Abbreviations: HSC, hepatic stellate cells; TEC, tubular epithelial cells; NK, natural killer cells; ROS, reactive oxygen species, Mtb, *M. tuberculosis*; CH25H, cholesterol 25-hydroxylase; FA, fatty acids; FASN, fatty acid synthase; PUFA, polyunsaturated fatty acids; SFA, saturated fatty acids; LXR, liver-X-receptor; RXR, retinoid-X-receptor; CLP, cecal ligation puncture; UPR, unfolded protein response (figure created with Biorender.com).

**Table 1 ijms-22-04076-t001:** Main features of CD5L during infection and sterile inflammation.

Role	Infection	Sterile Inflammation
**Pattern recognition receptor**	Binding to the following PAMPs: - LPS, LTA [18], PGN, mannan, β-D-glucan, zymosan [17]. - Binding and aggregation of Gram-positive and Gram-negative bacteria and fungi [17,18].	Binding to the following DAMPs: - Dead/necrotic cells and cellular debris [10]. - oxLDL [34].
**Apoptosis**	Inhibition of NKT and T-cell apoptosis in a mouse model of *Corynebacterium* infection [9]. Protection from bacterial-induced macrophage apoptosis [22,23,32,33,36].	Inhibition of dexamethasone-induced thymocyte apoptosis [8]. Inhibition of cycloheximide and CD96/FAS-induced apoptosis in J774A.1 macrophages [8]. Inhibition of macrophage apoptosis in a mouse model of atherosclerosis [43]. Inhibition of oxLDL-induced macrophage apoptosis in vitro [34,43]. Inhibition of hepatocyte apoptosis in IR injury in vitro [44].
**Uptake/Phagocytosis**	Increased uptake of latex beads [19], bacteria [9,21], and necrotic cells [41] by macrophages. No effect on the phagocytosis of microspheres, *E. coli* or *S. aureus* particles by macrophages [20].	KIM-1-dependent phagocytosis of dead cell debris by intraluminal cells in acute kidney injury [10]. Increased uptake of apoptotic hepatic cells by macrophages [20]. CD36-mediated oxLDL internalization by macrophages [34]. Decreased uptake of apoptotic neutrophils by macrophages [42].
**Microbiocidal**	Increased *L. monocytogenes* clearance by macrophages [22]. Inhibition of *M. tuberculosis* growth in macrophages through autophagy [23].	
**Inflammation**	Reduced macrophage secretion of TNF, IL-1β, and IL-6 and enhancement of IL-10 in response to PAMPs [17,18,20,26]. Increased ROS secretion by macrophages [18,23]. Restrains the proinflammatory module in non-pathogenic Th17 cells [28]. Proinflammatory role in mouse models of sepsis and acute lung injury [40,42]. Anti-inflammatory role in a mouse model of fungus-induced peritoneal injury [41].	Anti-inflammatory role in acute kidney injury [10], experimental autoimmune encephalomyelitis [28], and liver fibrosis [45]. Proinflammatory role in obesity [46], IgA nephropathy [47], and acute myocardial infarction [48].
**Ligands or Cell receptors**	CD36 in macrophages [26].	CD36 in macrophages and adipocytes [6,34,49]. Fatty acid synthase [6]. KIM-1 in kidney epithelial cells [10]. IgM pentamers [47,50,51]. IL-23 p19 [29].
**CD5L-producing cells**	Macrophages [19,20,21,22,23,32,41]. Non-pathogenic Th17 cells [28,29].	Macrophages [6,8,34,36,43,45]. Non-pathogenic Th17 cells [28,29]. Hepatocytes [45]. Retinal pigment epithelium [52]. Retinal microglial cells [52].

## Data Availability

The data presented in Figure 1 are openly available in the Uniprot Consortium web page (www.uniprot.org).

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
