# Peer review of "Multifaceted Roles of CD5L in Infectious and Sterile Inflammation"

_ijms, 2021, doi:10.3390/ijms22084076_

Round 1
Reviewer 1 Report
In the manuscript, Sanchez-Moral and coauthors comprehensively describe the state-of-the-art knowledge on the immune effector protein CD5-like molecule (CD5L). More particularly, the manuscript includes a review on both, well defined functions in anti-apoptosis, pattern recognition, cell polarization and lipid metabolism and its relevance in infection- and sterile-driven inflammatory conditions and metabolic/autoimmune diseases.
In summary, this is an interesting and well-performed compendium on basic and functional properties of CD5L that is further supported by a clearly arranged figure and a table. The manuscript comprises an extensive compilation of the most relevant references in the field and may help the reader to get easy access to current concepts of properties of CD5L, which are described in a comprehensible manner. There are no major issues regarding this review as the topic addressed is of scientific relevance and authors put high efforts in preparing the manuscript. There are, however, two minor issues as mentioned successive aiming to further improve the readability of the manuscript.
Minor points of improvement:
1. The review may further benefit from inclusion of a cartoon on the molecular characteristics/ domain structure of the CD5L protein.
2. Authors mostly introduced acronyms at their first appearance in the text but did not consequently performed that throughout the entire text. Thus, some additional abbreviations (e.g. LPS, LTA, PGN etc.) should be carefully introduced.
Author Response
Dear Reviewer,
We wish to thank you for your interest in our review. Below you will find a point-by-point answer to your comments:
1. We have included a new Figure showing a schematic diagram of mouse and human CD5L (Figure 1).
2. We apologize for the mistake. We have introduced the complete name for all abbrevations used throughout the text. These are written in red.
Reviewer 2 Report
Sarrias and co-authors provided an excellent review of CD5L progress in macrophages related inflammation and biology. The article presents smooth and beautiful writing.
Lines 155 and 214: ‘on the other hand’ may be replaced by “in addition”
Line 189: IL-23 is to drive and maintain the differentiation of Th17, rather than “expansion”
Author Response
Dear Reviewer,
We wish to thank you for your interest in our review. We have modified the text according to your suggestions, as you can see in red.